# In Pursuit of Causal Label Correlations for Multi-label Image Recognition

**Zhao-Min Chen[1],      Xin Jin[2],      Yisu Ge[1],\*      Sixian Chan[3]**

[1]Key Laboratory of Intelligent Informatics for Safety & Emergency of Zhejiang Province,
Wenzhou University
[2]Samsung Electronic (China) R&D Centre, Samsung Electronic
[3]The College of Computer Science and Technology, Zhejiang University of Technology
chenzhaomin123@gmail.com, ysg@wzu.edu.cn

## Abstract

Multi-label image recognition aims to predict all objects present in an input image. A common belief is that modeling the correlations between objects is beneficial for multi-label recognition. However, this belief has been recently challenged as label correlations may mislead the classifier in testing, due to the possible contextual bias in training. Accordingly, a few of recent works not only discarded label correlation modeling, but also advocated to remove contextual information for multi-label image recognition. This work explicitly explores label correlations for multi-label image recognition based on a principled causal intervention approach. With causal intervention, we pursue causal label correlations and suppress spurious label correlations, as the former tend to convey useful contextual cues while the later may mislead the classifier. Specifically, we decouple label-specific features with a Transformer decoder attached to the backbone network, and model the confounders which may give rise to spurious correlations by clustering spatial features of all training images. Based on label-specific features and confounders, we employ a cross-attention module to implement causal intervention, quantifying the causal correlations from all object categories to each predicted object category. Finally, we obtain image labels by combining the predictions from decoupled features and causal label correlations. Extensive experiments clearly validate the effectiveness of our approach for multi-label image recognition in both common and cross-dataset settings.

## 1 Introduction

Multi-label image recognition is a fundamental task in computer vision, aiming to predict all objects present in an image. It has widespread applications including object detection [6], medical imaging [39], and person re-identification [31]. However, this task is challenging as the combinations of labels can be tremendous. Modelling label correlations to reduce the search space is believed to be essential for multi-label image recognition [4].

In the research of multi-label image recognition, an implicit yet common assumption is: the training and test sets follow independent and identically distributions (i.i.d.), and the label correlations are consistent. Under this setting, a deep backbone network can implicitly extract context-aware features that are beneficial for object recognition, and furthermore, explicit label correlation modeling can explore contextual cues more deeply to improve the recognition accuracy. Technically, graph structures [4, 12] or attention mechanisms [10, 25] have been successfully employed to model label

---

\*Corresponding author.

38th Conference on Neural Information Processing Systems (NeurIPS 2024).

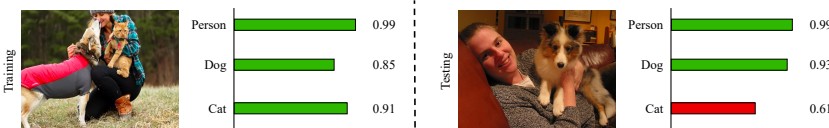

Figure 1: Illustration of the concept and effect of contextual bias in training. It is common that "Person", "Dog", and "Cat" co-occur in training images (we only show one image), while the test image may only contain "Person" and "Dog". Excessive reliance on the label co-occurrence in the training set may lead the recognition model to predict the "Cat" solely based on the presence of "Person" and "Dog".

correlations. However, these methods may fall short when there exists contextual bias in the training set. As illustrated by Fig. 1, "Person", "Dog", and "Cat" co-occur frequently in the training set, but a test image may only contain "Person" and "Dog". Consequently, due to the learned label correlations, the multi-label recognition model predicts a high probability of "Cat", solely based on the presence of "Person" and "Dog".

Recently, a few researchers [14, 17] have uncovered the contextual bias issue, and discarded explicit label correlation modeling. Furthermore, they attempted to alleviate the effects of contextual bias by decorrelating the feature representations of a category from its co-occurring context [14], or removing the contextual bias in features with causal mechanisms [17]. Despite improved accuracy, these methods neglect to model label correlations. In this paper, we ask: *is it possible to model label correlations for multi-label recognition, with the purpose of preserving good and suppressing bad contextual contents?* This work attempts to answer this question, as well as quantifying the goodness of contexts (corresponding to all pre-defined categories in the dataset) for recognizing a certain semantic category from the perspective of causal theory [22].

In this paper, we explore label correlations with a principled causal intervention approach for multi-label image recognition. Causal intervention aims to measure the causal effect from one cause variable to another effect variable, by "physically" putting the effect variable at any context to remove the effects of confounders in current image. Our key motivation lies on the realization that the causal label correlations (in the probability-raising sense [22]) are stable in both training and testing, even when contextual bias does exist in the training set. We pursue causal correlations (*e.g.*, from "Person" to "Clothes") to mine contextual cues for recognition, while suppressing spurious correlations (*e.g.*, from "Person" to "Cat") which are associated by confounders (*e.g.*, the overall scene) and may mislead the classifier in testing.

We design an end-to-end framework that carefully integrates causal intervention into multi-label recognition. Specifically, we decouple label-specific features with a Transformer decoder attached to the backbone network, and model the confounders (imaginary contextual contents) by clustering spatial features of all training images. Based on label-specific features and confounders, we employ a cross-attention module to implement causal intervention for all pre-defined categories, quantifying the causal label correlations from all object categories to each predicted object category. Finally, we combine the predictions from decoupled features and causal correlations for multi-label prediction.

In summary, our main contributions are as follows:

- We propose a conceptually simple, yet effective label correlation modeling approach based on causal intervention to tackle the issue of contextual bias for multi-label image recognition. It allows us to capture causal label correlations (good contextual contents) to improve recognition accuracy, while suppressing the effects of spurious correlations (possible bad contextual contents) that may mislead the classifier.

- We conduct comprehensive experiments with contextual bias to evaluate the effectiveness of the proposed method for multi-label image recognition. Under both common and cross-dataset settings, our method consistently demonstrates advantages over existing methods.

## 2   Related Work

**Correlation modeling for multi-label image recognition.**    Modeling label correlations is crucial for multi-label image recognition. Early approaches achieved this by embedding label correlations using

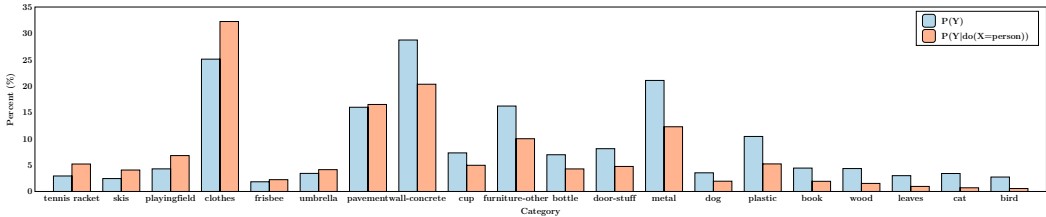

Figure 2: Illustration of causal label correlations and spurious correlations revealed by causal intervention, in a probability-raising sense that if $P(Y|do(X)) > P(Y)$, then a causal correlation exists from $X$ ("Person") to $Y$ (categories in this figure).

Recurrent Neural Networks (RNNs) [32, 21], but the performance is affected by the order in which the labels are predicted. To overcome the sequential issue, researchers attempted to capture label correlations with graph structures, enabling simultaneous prediction of the label sets [4, 28, 12, 25].

Besides graph structures, several other approaches have been developed to establish label correlations. DER [5] employs metric learning to pull related label-specific features closer and push unrelated label features apart. C-Tran [15] introduces a label mask training approach for general multi-label image classification, indirectly constructing label correlations by predicting masked labels. Q2L [19] employs Transformers to decompose label features and utilizes self-attention mechanisms to establish label correlations. SST [7] employs Transformers to simultaneously capture both spatial and semantic label correlations.

From a broader view of context modeling, label correlation modeling, arguably, can be understood as a strategy to enhance label-specific contextual cues. However, contextual information is a double-edged sword, and may mislead the classifier in presence of contextual bias in training.

**Contextual bias and debiasing.**   While visual context is widely believed to be beneficial for object recognition, recent works show that contextual bias may hurt multi-label image recognition [14, 17]. Such bias happens when an object category frequently co-occurs with some other object categories. Strongly relying on context may mislead the classifier, when typical contextual patterns around an object are absent or an object are absent from its typical context.

Due to the contextual bias issue, some recent works have discarded label correlation modeling for multi-label image recognition, since the contextual priors encoded by learnt label correlations may mislead recognition. Furthermore, these works developed contextual debiasing techniques, by decorrelating feature representations of a category from its co-occurring context [14], or removing contextual bias in features with causal mechanisms [17]. We argue that these methods may discard useful contextual cues, leading to inferior accuracy for common objects in common contexts.

**Causal intervention in vision.**   Causal intervention measures the causal relationship between two random variables, by removing the confounders that may associate them [22]. Recently, causal intervention has gained attention in the field of computer vision [17, 26, 33, 34, 27, 36, 37], with expectations to address the contextual bias or long-tailed distribution issues. For instance, VC-RCNN [33] employs causal intervention and proxy tasks to extract unbiased visual features, which can benefit downstream tasks like Image Captioning, Visual Question Answering, and Visual Commonsense Reasoning. Tang *et al*. [26] show that the SGD momentum is essentially a confounder in long-tailed classification, and propose to remove bad causal effects by intervention. Wang *et al*. [34] reveal that traditional attention module is biased in out-of-distribution setting, and propose causal attention for unbiased visual recognition.

For multi-label image recognition, Liu *et al*. [17] recently propose to remove the contextual bias in features with causal intervention. However, this approach does not consider label correlations, and may discard contextual evidences that are crucial for recognizing obscure instances.

## 3   Preliminaries and Motivation of Causal Correlations

**Causal correlations in probability-raising sense.**   Humans can easily understand the causal correlation between the presence of two objects (*e.g*., "Person" is the cause of "Clothes"). Seeking

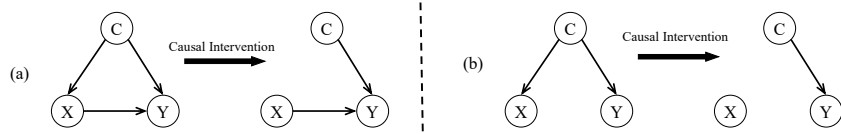

Figure 3: (a): Causal correlation between label $X$ and $Y$, which is not affected by confounder set $C$. (b): Spurious correlation, where the co-occurence of $X$ and $Y$ is caused by confounder set $C$.

for a calculable formal definition, we follow the statement in [23]: if $P(Y|do(X)) > P(Y)$, then a causal correlation exists from $X$ to $Y$ in a probability-raising sense. Here $do(X)$ is the $do$-operation, which pursues the causality between the cause $X$ and the effect $Y$ without the confounding effect.

As shown in Fig. 2, our implementation of causal intervention (which will be elaborated later) can reveal the causal correlations between objects in above probability-raising sense. For example, given the observation of "Person", the probabilities of "Clothes" and "Skis" raise. But on the other hand, although "Cat" often co-occurs with "Person", with causal intervention the probability of "Cat" even decreases when observing "Person".

**Why pursuing causal correlations?** Understanding this requires delving deep into the implication of causal intervention. Causal intervention is reminiscent of randomised controlled trials [2]. Based on the presence of $X$, causal intervention makes the prediction of $Y$ goes beyond the limitation of the context of current test image. It measures the probability of $Y$, by putting $Y$ at randomised context (which can be simulated by confounder set $C$) with $X$. Therefore, if $X$ is the cause of $Y$, it should be able to provide contextual cues to raise the probability of $Y$, regardless of the contents (except $X$ and $Y$) in current test image. If $X$ is not the cause of $Y$, it will not raise the probability of $Y$ based on causal intervention, although they might co-occur frequently in the training set.

Formally, Fig. 3 shows a Structural Causal Model [23], where $X$ and $Y$ represent two labels, and $C$ represents the confounder set. In causal theory, each directed edge denotes a possible causal relationship between two nodes. Fig. 3 (a) and (b) illustrate two extreme cases: the causal correlation between label $X$ and $Y$ which is not affected by the confounder $C$, and the spurious correlation, where the co-occurrence of $X$ and $Y$ is caused by the confounder set $C$.

**Causal intervention by backdoor adjustment.** As "physical" intervention that puts $Y$ at any context is almost impossible, backdoor adjustment [22] is typically applied for "virtual" intervention:

$$P(Y|do(X)) = \sum_c P(Y|X, C = c)P(C = c),$$
(1)

Here the key idea is to cut off the link from confounder $C$ to cause $X$, and stratify $C$ into pieces $C = \{c\}$, making $C$ no longer correlated with $X$, and making $X$ have a fair opportunity to incorporate every confounder $c$ into the prediction of $Y$, subject to a prior $P(c)$.

## 4 Approach

### 4.1 Overview of Proposed Pipeline

Building upon above analysis, we incorporate causal intervention into explicit label correlation modeling for multi-label image recognition, designing a pipeline of two complementary branches (Fig. 4): the branch of decoupled label-specific features, and the branch of causal label correlations. In particular, the causal correlation branch is built upon decoupled features.

Given an input image, a backbone network (*e.g.*, ResNet-50 [11]) is firstly employed to extract the spatial feature. Then, a Transformer decoder is leveraged to decouple label-specific features from the spatial feature. This branch predicts image labels based on objects themselves, rather than the context or label correlations. To take into account the label correlations yet overcoming the effects of contextual bias, we construct a causal intervention branch, which explicitly models causal label correlations, and integrates them into prediction.

Formally, we denote the prediction confidence from the causal intervention branch as $\hat{\boldsymbol{y}}_{causal}$, and the prediction from the decoupled feature branch as $\hat{\boldsymbol{y}}_{decouple}$, then the final prediction confidence

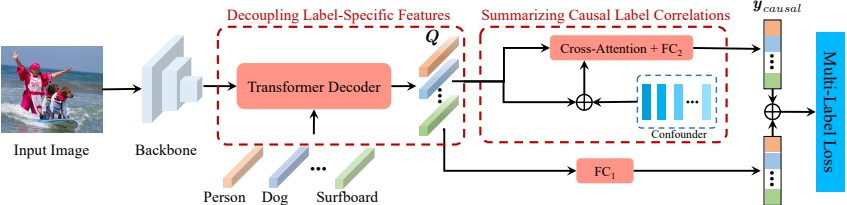
Figure 4: The overall framework of our proposed method.

scores $\hat{\boldsymbol{y}}$ can be written as:

$$\hat{\boldsymbol{y}} = 1/2 \cdot \hat{\boldsymbol{y}}_{causal} + 1/2 \cdot \hat{\boldsymbol{y}}_{decouple} \in \mathbb{R}^N \,, \tag{2}$$

where $N$ is the number of categories. We utilize standard multi-label recognition loss to train the model, which can be written as:

$$\mathcal{L} = \sum \boldsymbol{y}_{gt} \log(\hat{\boldsymbol{y}}) + (1 - \boldsymbol{y}_{gt}) \log(1 - \hat{\boldsymbol{y}}) \,, \tag{3}$$

where $\boldsymbol{y}_{gt} = \{0, 1\}^N$ is ground truth label vector of the input image.

In the following, we will detail the designs of the decoupled feature branch and the causal intervention branch. We will also present explanations and discussions about causal correlations in context to facilitate understanding.

## 4.2   Predicting by Decoupling Label-Specific Features

We firstly decouple label-specific features for input image with two purposes: (i) predicting image labels based on objects themselves; (ii) preparing for label correlation modeling by causal intervention.

For decoupling label-specific features, common approaches include Class Activation Mapping (CAM) [40] and Transformer [30]. We employ a Transformer decoder for this purpose. Specifically, given input image $\boldsymbol{I}$, we firstly use a CNN backbone to extract spatial feature $\boldsymbol{F}$:

$$\boldsymbol{F} = f_{cnn}(\boldsymbol{I}). \tag{4}$$

Then, we employ a standard Transformer decoder to decouple label-specific features $\boldsymbol{X} \in \mathbb{R}^{N \times D}$ from $\boldsymbol{F}$:

$$\boldsymbol{X} = f_{decoder}(\boldsymbol{Q}, \boldsymbol{F}). \tag{5}$$

Here, $\boldsymbol{Q} \in \mathbb{R}^{N \times D}$ are learnable label embedding as queries, $N$ and $D$ are the number of categories and dimensionality of spatial features, respectively.

Finally, we obtain the prediction confidence $\hat{\boldsymbol{y}}_{decouple}$ from decoupled features by:

$$\hat{\boldsymbol{y}}_{decouple} = \sigma(f_{fc1}(\boldsymbol{Q})) \,, \tag{6}$$

where $f_{fc1}(\cdot)$ denotes the fully-connected layer, $\sigma(\cdot)$ is the sigmoid function. We apply multi-label loss to this branch for decoupled label-specific feature learning.

## 4.3   Predicting by Summarizing Causal Label Correlations

With label-specific features, we can construct our intervention branch which *explicitly* models causal label correlations for multi-label image recognition.

To estimate the probability of each category $Y_j$ on this causal intervention branch, a straightforward approach is firstly calculating $P(Y_j|do(X_i))$ for each category $X_i$, and then merging them:

$$\hat{y}^j_{causal} = f_{merge}([P(Y_j|do(X_1)), ..., P(Y_j|do(X_N))]). \tag{7}$$

One might expect complex modeling of $P(Y_j|do(X_i))$ and $f_{merge}(\cdot)$ for calculating Eq. 7. In this work, we introduce a simple yet effective implementation, using one cross-attention layer and one fully-connected layer.

**Causal intervention based on label-specific features.** We hypothesize that whether $X_i$ appears in the test image should be encoded by its label-specific feature $x_i$. Given label-specific features $x_i$ and $y_j$ for label $X_i$ and $Y_j$, and one potential confounder feature $c$, we model the conditional likelihood in Eq.1 as:

$$P(Y_j|X_i, C = c) = \sigma(f_{y_j}(x_i, c)), \tag{8}$$

where $\sigma(\cdot)$ is the sigmoid function. $f_{y_j}(\cdot)$ calculates the logit for label $Y_j$. Then, causal intervention can be calculated as:

$$P(Y_j|do(X_i)) = \mathbb{E}_c[\sigma(f_{y_j}(x_i, c))]. \tag{9}$$

Due to the difficulty in directly computing Eq. 9, we apply the Normalized Weighted Geometric Mean (NWGM) [13] to approximate the above equation:

$$\begin{aligned} P(Y_j|do(X_i)) &= \mathbb{E}_c[\sigma(f_{y_j}(x_i, c))] \\ &\approx \sigma(\mathbb{E}_c[f_{y_j}(x_i, c)]) \\ &= \sigma(\sum_c f_{y_j}(x_i, c) \cdot P(c)). \end{aligned} \tag{10}$$

Here $P(c)$ is the prior of confounder, which can be obtained from data.

Calcualting Eq. 10 requires further modeling of $f_{y_j}(x_i, c)$. However, instead of hypothesizing a formulation of $f_{y_j}(x_i, c)$ which only intervenes one category $X_i$ for $Y_j$, we describe an efficient formulation based on cross attention to intervene all categories $X_i$ for $Y_j$ in one step. This allows us to circumvent the calculation of each $f_{y_j}(x_i, c)$, making the calculation of Eq. 7 more efficient.

**Effective modeling for all $f_{y_j}(x_i, c)$ by cross-attention.** Formally, following the notation in Eq. 5, let $\boldsymbol{X} = [x_1, ..., x_N] \in \mathbb{R}^{N \times D}$ denote all label-specific features. To implement Eq. 7, we seek for a model to combine the information of all label-specific features $x_i$ and one potential confounder feature $c$ to predict the logit of label $Y_j$. We employ cross-attention mechanism and fully-connected layer for this purpose:

$$\begin{aligned} \boldsymbol{Z}_c &= \boldsymbol{X} + c, \\ \hat{y}^j_{causal} &= f_{merge}([P(Y_j|do(X_1)), ..., P(Y_j|do(X_N))]) \\ &= f_{merge}([\sigma(\sum_c f_{y_j}(x_1, c) \cdot P(c)), ..., \sigma(\sum_c f_{y_j}(x_N, c) \cdot P(c))]) \\ &\approx \sigma(\sum_c f_{y_j}(\boldsymbol{X}, c) \cdot P(c)) \\ &= \sigma(\sum_c f_{fc2}(f_{cross\_atten}(y_j, \boldsymbol{Z}_c, \boldsymbol{Z}_c)) \cdot P(c)), \end{aligned} \tag{11}$$

where $\boldsymbol{Z}_c$ is the addition-based combination of all label-specific features $\boldsymbol{X}$ and a confounder feature $c$, and $f_{fc2}$ is a fully-connected layer applied upon the cross-attention feature to obtain the logit. Algorithm 1 provides the pseudocode of causal intervention process.

**Modeling the confounders** It remains an open question about cofounder modeling for visual recognition tasks. In VC R-CNN [33], the authors treated objects as confounders, and extract object-level features based on bounding box annotations. However, on one hand, there is no bounding box annotation in the typical setting for multi-label image recognition. On the other hand, we argue that cofounders for recognizing certain object are often hard to define and enumerate – objects, scene, and even the texture of the environment are all potential confounders. For example, suppose that an image contains two objects: "`Person`" and "`Surfboard`", with the scene being the "`Beach`". In VC R-CNN, the "`Surfboard`" is considered the confounders for the "`Person`". However, in our opinion, the true confounder should the "`Beach`", although it does not have an associated image label.

Based on above analysis, to characterize these non-enumerable confounders, high-level spatial features of training images from pre-trained classification CNN provide a good choice, as semantic objects/regions are often activated in classification features. By clustering spatial features with K-means algorithm, we obtain a compact set of prototypes to represent potential confounders like objects, scenes and textures. We empirically show the effectiveness of this simple approach for modeling the confounders.

**Algorithm 1** Pseudocode of Causal Intervention in a PyTorch-like style.

```
# X (NxD): The label-specific features
# C (MxD): Confounders
# P (M): Priors
# N is the number of categories
# D is the dimensionality of spatial features
# M is the number of confounders
Z = X.unsequeeze(1) + C.unsequeeze(0) # NxMxD
y_j = X[j,:]
y_causal_j = 0
for c in range(M):
    y_causal_j += fc2(cross_atten(y_j, Z[:,c,:], Z[:,c,:])) * P[c]
y_causal_j = y_causal_j.sigmoid()
```

## 5 Experiments

### 5.1 Experimental Settings and Implementation Details

**Common Setting.** The COCO-Stuff [1] and DeepFashion [20] datasets are experimented in common setting, where the training and test sets are from the same dataset. We strictly adhere to the evaluation setup employed in [14], and report the performance under two different test distributions: "*Exclusive*" denotes virtual co-occurrence where labels appearing simultaneously in the training set do not co-occur in the test set, and "*Co-occur*" represents the objects co-occurring in both the training and test sets. "*All*" is the average performance of all categories. We report top-3 recall for DeepFashion and mAP for COCO-Stuff.

**Cross-dataset Setting.** We consider a more challenging yet practical setting in real-world applications: the training and test sets are from different datasets, and may suffer from serious contextual bias issue. We simulate this setting by using MS-COCO [16] for training and NUS-WIDE [8] for testing, and vice versa. In particular, we select the same categories (14 common classes) from both the MS-COCO and NUS-WIDE datasets for experiments. We report the mean Average Precision (mAP) for all categories in this setting.

**Implementation Details.** To fair comparison with previous methods, we employ ResNet-50 and ResNet-101 as backbones for the common setting and real-world setting, respectively. We utilize the ImageNet [9] for model parameter initialization. For the cross-dataset setting, the input images are randomly cropped to a resolution of $448 \times 448$, while for the common setting, the resolution is set to $224 \times 224$. To extract label-specific features, we employ a 2-layer Transformer decoder with 4 attention heads. For modeling confounders, we first train a baseline model with standard multi-label loss, then extract spatial features from all the images in the training dataset, and employ $K$-means algorithm (default number of clusters is set to 80) to cluster all pixel-level spatial features. The Adam optimizer is chosen for model optimization, with a weight decay of $2e-2$ and $(\beta_1, \beta_2) = (0.9, 0.9999)$. The initial learning rate is set to $1e-4$, and we employ a cyclic learning rate policy to train our model for 80 epochs. All of the experiments are run on a computer with an AMD EPYC 7542 32-Core processor, 256 GB main memory, and eight GTX-3090 GPUs.

### 5.2 Comparing to the State-of-the-arts

**Common Setting.** In Table 1, we report the performance on COCO-Stuff and DeepFashion datasets, where the baseline is the vanilla Resnet-50 with standard multi-label loss. Comparing with the baseline, we observed a significant improvement ($55.0\%$ *v.s.* $60.6\%$ on "*All*" mAP). Furthermore, our proposed method outperforms all state-of-the-arts on these two benchmarks. For example, it obtains a $+1.5\%$ "*Exclusive*" and $+3.6\%$ "*Co-occur*" improvements over the feature-split [14] on COCO-Stuff dataset.

We also observe that previous methods that directly build label correlations based on graph structures are affected by contextual bias in datasets. For example, ML-GCN [4] and SSGRL [28] both construct label graphs based on the training set and utilize the graph structure to capture correlations between

Table 1: Performance comparison in the common setting on the COCO-Stuff and DeepFashion datasets.

| Method | COCO-Stuff (mAP) | | | Deepfashion (top-3 recall) | | |
|---|---|---|---|---|---|---|
| | Exclusive | Co-occur | All | Exclusive | Co-occur | All |
| Q2L [19] | 23.5 | 67.1 | 57.2 | 12.8 | 26.3 | 26.1 |
| ADD-GCN [12] | 20.6 | 64.8 | 55.2 | 8.2 | 22.6 | 23.5 |
| ML-GCN [4] | 18.6 | 67.1 | 55.1 | 10.3 | 23.7 | 24.0 |
| SSGRL [28] | 18.1 | 66.6 | 54.9 | 7.9 | 22.8 | 23.1 |
| C-Tran [15] | 22.4 | 65.1 | 55.4 | 11.4 | 24.6 | 24.8 |
| CCD [17] | 23.8 | 65.3 | 55.9 | 11.5 | 24.2 | 24.6 |
| TDRG [38] | 20.0 | 64.8 | 56.2 | 8.1 | 22.9 | 23.6 |
| IDA [18] | 25.2 | 64.9 | 57.0 | 11.3 | 25.1 | 25.4 |
| CAM-Based [14] | 26.4 | 64.9 | – | – | – | – |
| feature-split [14] | 28.8 | 66.0 | – | 9.2 | 20.1 | – |
| Baseline (R50) | 21.9 | 65.5 | 55.0 | 11.5 | 24.1 | 24.1 |
| Ours | **29.7** | **69.6** | **60.6** | **14.6** | **27.4** | **28.8** |

Table 2: mAP Performance comparison in the cross-dataset setting on the MS-COCO and NUS-WIDE datasets.

| Method | MS-COCO → NUS-WIDE | NUS-WIDE → MS-COCO |
|---|---|---|
| ADD-GCN [12] | 81.8 | 77.2 |
| ML-GCN [4] | 81.4 | 77.2 |
| SSGRL [28] | 80.2 | 76.1 |
| C-Tran [15] | 80.9 | 76.9 |
| CCD [17] | 81.9 | 78.3 |
| Q2L [19] | 82.1 | 78.6 |
| IDA [18] | 82.3 | 78.9 |
| CAM-Based [14] | 81.0 | 77.8 |
| feature-split [14] | 81.9 | 78.3 |
| Baseline (R101) | 81.1 | 77.1 |
| Ours | **83.2** | **80.2** |

labels. These two methods cannot achieve significant improvements over the baseline, and even exhibit noticeable performance degradation. We speculate that the graph structure constructed from the training set are not applicable to the test set in presence of contextual bias.

**Cross-dataset Setting.** Table 2 reports the results on the MS-COCO and NUS-WIDE datasets, where the baseline is the vanilla Resnet-101 with standard multi-label loss. Similar to the common setting, the performance of previous methods is affected by contextual bias, and our method outperforms all other state-of-the-art methods. However, compared to the baseline, the performance improvement of our method is not as significant as in the common setting. We speculate that, apart from contextual bias in training, the inconsistency in data distribution may also affect the performance. Even so, the cross-dataset results can indicate robustness and generalization capabilities of our method.

## 5.3 Ablation Studies

In this section, we conduct ablation studies by using ResNet-50 as backbone on COCO-Stuff Dataset.

### 5.3.1 Effectiveness of Different Modules

We investigate the impacts of key modules in our framework. Specifically, there are two essential modules, *i.e.*, decoupling the label feature module (denoted as "Decouple") and causal intervention module (denoted as "Causal"). Table 3 shows the mAP performance by progressively integrating the above two modules. Solely applying "Decouple" on the backbone gives a +1.8% "*All*" mAP, +0.2% "*Exclusive*" mAP and +1.5% "*Co-occur*" mAP improvement. Directly applying decoupling leads to improved "*Co-occur*" performance but fails to enhance "*Exclusive*" performance. We speculate

Table 3: The impacts of different modules.

| Decouple | Causal | Exclusive | Co-occur | All |
|---|---|---|---|---|
| | | 21.9 | 65.5 | 55.0 |
| √ | | 22.1 | 67.0 | 56.8 |
| √ | √ | **29.7** | **69.6** | **60.6** |

Table 4: The impacts of clustering center number.

| Number | Exclusive | Co-occur | All |
|---|---|---|---|
| 20 | 26.9 | 68.9 | 59.6 |
| 40 | 26.8 | 69.3 | 60.1 |
| 60 | 29.3 | **69.8** | 60.2 |
| 80 | **29.7** | 69.6 | **60.6** |
| 100 | 29.5 | 69.4 | 60.5 |

Table 5: The impact of backbones for clustering.

| Confounder Backbone | Exclusive | Co-occur | All |
|---|---|---|---|
| ResNet-50 | **29.7** | 69.6 | **60.6** |
| ResNet-101 | 29.6 | **69.9** | 60.5 |
| BEIT3-Large | 29.4 | 69.7 | 60.5 |

Table 6: The impact of different modeling approaches for confounders.

| Method | Exclusive | Co-occur | All |
|---|---|---|---|
| Random | 22.1 | 66.3 | 56.1 |
| Early | 27.8 | 69.1 | 60.1 |
| Label | 28.0 | 69.3 | 60.3 |
| $K$-means | **29.7** | **69.6** | **60.6** |

that the cross-attention mechanism can capture the long-range dependencies, since it can implicitly model causal correlations but cannot eliminate spurious co-occurrence correlations. Then, the causal intervention module, which can remove spurious co-occurrence correlations and capture the causal correlations, bring another $3.8\%$ "*All*" mAP. These results show the effectiveness of our approach in alleviating label correlations bias.

### 5.3.2 Investigation of Confounders

Modeling confounders is a core step in our approach. In order to investigate the impact of different confounders, we designed comparative experiments at three aspects, *i.e.*, the number of clustering centers, different clustering features, and different approaches for modeling confounders.

**The number of clustering centers.** In order to investigate the influence of the number of clustering centers, we conducted experiments with different numbers of cluster centers: 20, 40, 60, 80, and 100, respectively. The experimental results are presented in the Table 4. It can be observed that within a certain range, increasing the number of cluster centers does not significantly affect performance. However, a performance decline is noticeable when the number of clusters is reduced to 20. We speculate that the confounders correspond to many attributes including semantic, color, texture, and so on. A smaller number of cluster centers can only express very limited attributes, resulting in a decline in performance.

**Different backbones for clustering.** By default, we employ ResNet-50 as the backbone to extract features from all images in the training set on COCO-Stuff, and perform clustering to obtain confounders. In order to investigate the impact of different clustering features, we utilize distinct backbones for feature extraction. As shown in Table 5, the confounders obtained from different backbones have not significant influence on final performance. The role of the confounders is to identify label causal correlations and does not directly participate in recognition. Therefore, the features extracted by a weak backbone seems sufficient in identifying causal correlations, and the use of a strong backbone does not lead to performance improvement.

**Different approaches for modeling confounders.** In this paper, we utilize clustering centers to characterize confounders. We compare it with two additional modeling techniques. The experimental results as shown in the Table 6. "Random" means using random vectors to replace cluster centers as confounders. "Early" means fusing features from early epochs directly to obtain the confounder, which is adopted by [17]. "Label" means directly using label-specific features as the confounder. Our modeling of confounders using $K$-means yields the best results. "Random" leads to significant performance degradation (measured by mAP), especially on the "Exclusive" subset. Since, we believe that confounders should be modeled at semantic level. The "Early" approach relies solely on simple feature fusion, which fails to effectively differentiate various attributes. The "Label" approach only employs object semantics as the confounder, leading to the omission of other attributes. By contrast,

Table 7: Different implementations of Eq. 7.

| Method | Exclusive | Co-occur | All |
|--------|-----------|----------|------|
| Linear | 27.4 | 69.1 | 60.0 |
| Ours | **29.7** | **69.6** | **60.6** |

our approach incorporates both object-related information and other contextual information, offering a more expressive set of confounders.

### 5.3.3 Different Implementations of Eq. 7

We investigate different implementations of Eq. 7. Following [33], we employ a simpler linear approach and average merge method to model Eq. 7. As shown in Table 7, we can observe that employing a linear modeling approach yields competitive results, but our method still outperforms it. We speculate that while linear modeling can implement Eq. 7, our approach utilizes cross-attention, which possesses the ability to model long-range dependencies, thereby efficiently capturing causal correlations and suppressing spurious correlations.

## 6 Conclusions

In this paper, we presented a principled approach to address the contextual bias issue for multi-label image recognition. Using causal intervention from causal theory, we pursued causal label correlations, and integrated them into multi-label prediction. We evaluate the effectiveness of our approach with both quantitative and qualitative assessments. In the future, we will investigate more advanced structural causal models for better describing the label correlations.

**Acknowledgements**
This research was supported by the National Natural Science Foundation of China (Grant No. 62202337, 61906168, 62072340), in part by the Zhejiang Provincial Natural Science Foundation of China (Grant No. LQ24F020016, LY23F020023), Key Science and Technology Innovation Project of Wenzhou(Grant No. ZG2022014), and in part by the Wenzhou Key Laboratory Construction Project (Grant No. 2022HZSY0048).

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

# A    Supplemental Material

## A.1    Limitations

Despite improved accuracy, our causal intervention-based multi-label image recognition algorithm still has several limitations. First, we model confounders by clustering the spatial features extracted from a pre-trained classification CNN. On one hand, this approach is dataset-dependent, and on the other hand, although our modeled confounders mainly concerne with object-level, texture-level and scene-level concepts, it is not possible to determine the specific semantics of these concepts, resulting in limited interpretability. Second, we only integrate the causal intervention technique in causal theory in our multi-label image recognition pipeline. In the future, we will consider higher-level causality such as counterfactual reasoning.

## A.2    Experiments

### A.2.1    Datasets

COCO-Stuff [1] covers 171 classes and contains 82,783 training images and 40,504 testing images. DeepFashion [20] contains 209,222 training images and 40,000 testing images. Following [14], for DeepFashion we only consider the 250 most frequent attributes in the training data since other attributes do not contain sufficient samples.

We also evaluate our method in more challenging cross-dataset setting: training on MS-COCO [16], but testing on NUS-WIDE [8] dataset. The MS-COCO dataset contains 122,218 images and covers 80 common objects, with an average of 2.9 labels per image. The NUS-WIDE dataset has 269,648 images and 81 concepts, with an average of 2.4 concept labels per image.

### A.2.2    Intra-dataset Comparisons on MS-COCO

In our main paper, we present cross-dataset experiments to show the effectiveness of our method for contextual bias and distribution shift. That is, we train on MS-COCO [16], but test on NUS-WIDE [8]; or train on NUS-WIDE [8], but test on MS-COCO [16]. Here, we also report the accuracy of our method for general multi-label recognition, by both training and testing on MS-COCO.

As shown in Tab.8, although our approach is designed to solve the contextual bias issue for multi-label recognition, it can achieve competitive accuracy for general multi-label recognition. In particular, our method outperform CCD [17] by 0.9 mAP, which also integrates causal mechanisms for contextual de-biasing, but discards the explicit modeling for label correlations. Combining the cross-dataset and intra-dataset experimental results, our approach shows advantages over both general multi-label image recognition algorithms [7, 19] and recent algorithms that explicitly considers contextual de-biasing [14, 17].

### A.2.3    Effect of different backbones.

To investigate the impact of different backbones on performance, we conducted experiments using ResNet-50, ResNet-101, and BEIT-Large [35] as backbones. The results are shown in Table 9, which reveal that even with a stronger backbone, our method can still effectively improve performance. For example, our method with the BEiT-Large backbone achieves a 72.2% mAP, which outperforms the baseline by 2.9% mAP. This experimental result demonstrates that our method can be generalized to stronger backbones.

### A.2.4    Visualizations

For qualitative verification, we employ Grad-CAM [24] to visualize label-specific features for the baseline and our proposed method. As shown in Fig. 5, although the baseline can activate relevant object locations, it also activates the regions with spurious correlations for the target object. In contrast, our method activates solely at the object's location. In Fig. 5 (a), besides activating label "Cell Phone", the baseline method additionally activates label "Person", whereas our method solely activates "cell phone." Regarding negative labels, when an input image lacks a specific label, the baseline might erroneously infer false labels through other labels due to excessive reliance on label co-occurrences. Our method mitigates this phenomenon effectively. In Fig. 5, as labels

Table 8: Comparisons with state-of-the-art on MS-COCO [16], where the models are all trained on MS-COCO dataset. Here, * means the re-produced results by ourselves.

| Methods | Resolution | mAP |
|---|---|---|
| CNN-RNN [32] | 224×224 | 61.2 |
| SRN [41] | 224×224 | 77.1 |
| ResNet-101* [11] | 448×448 | 79.1 |
| SSGRL* [28] | 448×448 | 81.9 |
| DER [5] | 448×448 | 82.8 |
| ADD-GCN* [12] | 448×448 | 82.8 |
| ML-GCN [4] | 448×448 | 83.0 |
| C-Tran* [15] | 448×448 | 83.1 |
| P-GCN [3] | 448×448 | 83.2 |
| MCAR [10] | 448×448 | 83.8 |
| MS-CMA [25] | 448×448 | 83.8 |
| CCD [17] | 448×448 | 84.0 |
| SST [7] | 448×448 | 84.2 |
| TDRG [38] | 448×448 | 84.6 |
| IDA [18] | 448×448 | 84.8 |
| Q2L [19] | 448×448 | **84.9** |
| Ours | 448×448 | **84.9** |

Table 9: The impacts of different backbones on COCO-Stuff.

| Backbone | Ours | Exclusive | Co-occur | All |
|---|---|---|---|---|
| ResNet-50 |  | 21.9 | 65.5 | 55.0 |
|  | √ | **29.7** | **69.6** | **60.6** |
| ResNet-101 |  | 23.7 | 68.5 | 58.9 |
|  | √ | **31.0** | **70.3** | **61.4** |
| BEIT3-Large |  | 46.8 | 80.4 | 69.3 |
|  | √ | **56.6** | **81.2** | **72.2** |

"Person", "Baseball Glove", and "Baseball Bat" co-occur in the training set, the baseline infers the presence of "Baseball Bat" through "Person" and "Baseball Glove", even if it is absent in the input image. This eventually leads the baseline to incorrectly activate spatial features associated with the label "Baseball Bat". In contrast, our method nearly avoids activating such features. The visualization results further validate our motivation, *i.e.*, our approach effectively removes spurious co-occurrence correlations and captures the causal correlations.

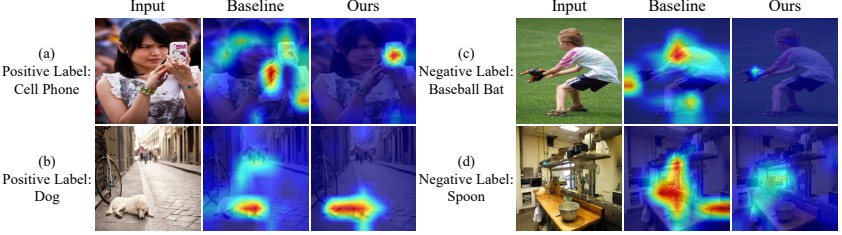

Figure 5: Visualization for the spatial feature map of baseline and our proposed method using the Grad-CAM [24].

### A.2.5  *t*-SNE Visualization of Features

In Fig.6, we show the *t*-SNE [29] visualization of our confounders and two typical label-specific features ("Baseball Glove" and "Person").

Without causal intervention, we observe that "Baseball Glove" and "Person" are mixed with each other, indicating that they often co-occur in the training set. Furthermore, the confounders are around

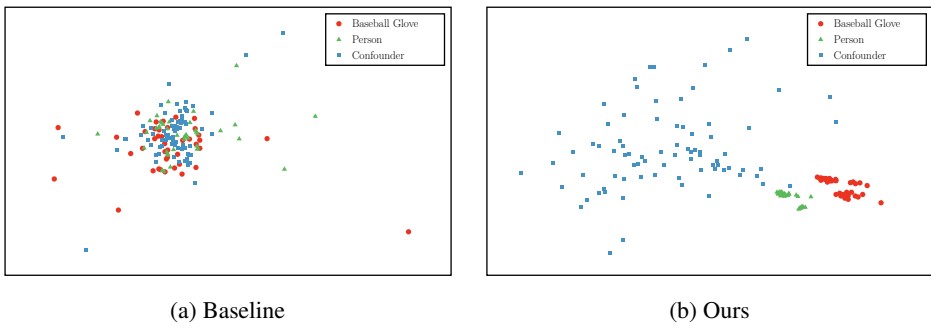

(a) Baseline        (b) Ours

Figure 6: $t$-SNE [29] visualization for confounders and label-specific features of "`Baseball Glove`" and "`Person`".

label-specific features, since they are obtained by clustering the spatial features of a pre-trained classification backbone.

With our causal intervention branch, we empirically find that "`Baseball Glove`" and "`Person`" are pulled apart, since they have very weak causal correlations in the probability-raising sense. On the other hand, the label-specific features are pulled away from the confounders, suggesting that our causal intervention can remove the influence of confounders for causal representation learning.

