# OpenReview forum: "In Pursuit of Causal Label Correlations for Multi-label Image Recognition"
_NeurIPS.cc/2024/Conference — NeurIPS 2024 poster_

### Official Review · Reviewer_BNe7 · 2024-06-19

**Soundness:** 4
**Presentation:** 4
**Contribution:** 4
**Rating:** 8
**Confidence:** 5

**Summary:**

This paper proposes a simple yet effective method based to address the issue of contextual bias for multi-label image recognition. It utilizes the casual intervention theory to pursue causal label correlations and suppress spurious label correlations. It utilizes the k-means to model the confounders, and employs the cross-attention mechanism to achieve the causal intervention. Experimental results demonstrate the efficacy of this approach.

**Strengths:**

The paper is well-written and easy to understand.

This method seems easy to implement.

The approach achieves good results.

The problem is interesting in multi-label recognition tasks.

**Weaknesses:**

Why the k-means algorithm is used to build confounders, the author should give further explanation.

In the paper, the number of cluster centers is only calculated to 100, and what will happen if it continues to increase?

Regarding inference time, how many forward passes does the method require?

In L191, how to obtain P(c) from the data?

**Questions:**

The authors should provide more detailed explanations and experiments about confounders.

The authors should provide a description of the inference process.

The authors should clarify how to obtain a prior of confounders.

**Limitations:**

The authors have addressed the limitation of this method. This method is dataset-dependent, and it can not determine the specific semantics of confounders, resulting in limited interpretability. Besides, it only considers the causal intervention, and does not consider causal reasoning.

---

> ### Author Rebuttal · Authors · 2024-08-05
>
> We thank reviewer BNe7 for the positive comments on our work. In the following, we present our responses addressing the raised concerns.
>
> **(Weakness 1)** In this work, we apply K-means clustering on the spatial features extracted from a pre-trained classification network for confounder modeling. This approach is based on our realization that the cofounders for recognizing certain object are often hard to define and enumerate – objects, scene, and even the texture of the environment are all potential confounders. On the other hand, a pre-trained classification CNN tends to activate the discriminative regions (which could be objects, scene and even the texture) in an image. Therefore, K-means clustering on CNN spatial features can produce a compact set of prototypes to represent potential confounders like objects, scenes and textures.
>
> **(Weakness 2)** Thanks for your suggestion. We conduct additional experiments regarding the number of the clustering centers, as illustrated in Fig. 2 in our attached pdf file. Our experiments show that 80 clusters are sufficient to represent potential confounders in our datasets, and 400 clusters does not bring significant accuracy gain. We point out that this conclusion still remains as an empirical observation. And, we hypothesize that for more complex multi-label image classification datasets (which might be collected in the future), further increasing the number of clusters may be necessary for modeling complex confounders and achieving higher accuracy.
>
> **(Weakness 3)** As the confounders are pre-computed, only one forward pass is required to obtain the final prediction.
>
> **(Weakness 4)** With K-means clustering, each spatial feature will be assigned to a certain cluster $c$. We calculate the number of spatial features for each cluster, divide it by the total number of spatial features, and then obtain $P(c)$ for each cluster.
>
> **(Question 1)** Thanks for your suggestion. We plan to add more explanations about confounders (e.g., our response to Weakness 1) and more experiments (e.g., other implementation choice for confounders like random vectors) in our revised version.
>
> **(Question 2)** Since the confounders (and the prior $P(c)$) are pre-computed, the inference process of the causal branch (formulated by Eq. 11) only requires one forward pass. As a result, the inference process of our whole pipeline (Eq. 2) also only requires one forward pass. We will further detail the inference process of our method in the revised version.
>
> **(Question 3)** Please refer to our response to Weakness 4.
>
> **(Limitations)** As described in Sec A.1, our current approach still has limitations in interpretability and higher-level causal modeling. We consider these as our future works, and will try our best to solve these limitations.

---

> > ### Comment · Reviewer_BNe7 · 2024-08-12
> > **Respond to the authors**
> >
> > The authors have addressed my concerns, I will raise my score correspondingly.

---

> ### Author Response · Authors · 2024-08-12
> **Response to Reviewer BNe7**
>
> We are thankful for your acceptance and constructive feedback.

---

### Official Review · Reviewer_wRCW · 2024-07-10

**Soundness:** 3
**Presentation:** 3
**Contribution:** 3
**Rating:** 4
**Confidence:** 3

**Summary:**

This paper presents a novel approach to addressing label correlations in multi-label image recognition by using causal intervention. The method involves decoupling features, modeling confounders, and implementing causal interventions to capture useful contextual information while suppressing spurious label correlations. This approach is highly innovative and has significant potential applications.

**Strengths:**

1. **Innovative Approach:**
   The paper introduces a novel method that applies causal intervention to model label correlations in multi-label image recognition. This innovative approach addresses the challenge of spurious label correlations and captures useful contextual information, which is a significant advancement in the field.

2. **Comprehensive Methodology:**
   The proposed framework integrates several complementary techniques, including feature decoupling with a Transformer decoder, confounder modeling through clustering, and causal intervention using cross-attention mechanisms. This comprehensive methodology enhances the robustness and accuracy of multi-label image recognition models.

3. **Thorough Experimental Validation:**
   The paper conducts extensive experiments across multiple datasets, demonstrating the effectiveness of the proposed method. The results consistently show improvements over existing approaches, particularly in scenarios with contextual biases, underscoring the practical value of the method.

**Weaknesses:**

1. **Lack of Hyperparameter Analysis:**
   The paper does not provide a detailed analysis of the hyperparameters involved in the proposed method, such as the number of clusters for confounders or the parameters of the cross-attention module. A sensitivity analysis of these hyperparameters would be beneficial to understand their impact on model performance and to guide practitioners in tuning the model effectively.

2. **Insufficient Discussion on Method Limitations:**
   The paper lacks a thorough discussion on the limitations of the proposed method. It would be valuable to include an analysis of scenarios where the method might not perform well, such as when the selection of confounders is inaccurate or when the causal relationships between labels are weak. Addressing these limitations can provide a more balanced view of the method's applicability and robustness.

3. **Limited Ablation Studies:**
   Although the paper includes some ablation studies, the number and depth of these experiments are not comprehensive enough. More detailed ablation studies are needed to analyze the independent contribution of each module (e.g., feature decoupling, confounder modeling, and causal intervention) to the overall performance. This would help in understanding the importance and effectiveness of each component of the proposed method.

**Questions:**

1.  I noticed another paper titled "Counterfactual Reasoning for Multi-Label Image Classification via Patching-Based Training" that also employs causal inference to address multi-label image classification. The methods in these papers differ in implementation and theoretical basis. Could you further elaborate on the main differences and advantages of your approach compared to this work?
2.  Your paper does not provide a detailed analysis of the hyperparameters involved in the proposed method. Could you explain the rationale behind the chosen hyperparameters and their impact on the model's performance?
3. There is a lack of discussion on the limitations of your proposed method. In what scenarios might your method underperform, and how could future work address these limitations?
4. Could you explain why certain confounders were selected for modeling in your approach? How does the choice of confounders impact the effectiveness of causal intervention in your model?
5. How does your method handle cases where the causal relationships between labels are weak or not well-defined? Does this affect the model's accuracy, and if so, how?
6. How does your approach ensure robustness against noise and variability in the data? Are there any specific strategies employed to handle noisy or incomplete labels?
7. Could you provide more details on how the feature decoupling using the Transformer decoder specifically contributes to reducing contextual biases in multi-label image recognition?

**Limitations:**

The effectiveness of the proposed method relies heavily on accurately modeling the confounders. If the selection of confounders is not precise or representative of the underlying data distribution, the causal intervention may not effectively distinguish between useful contextual information and spurious correlations. This could potentially limit the method's performance in scenarios where confounder selection is challenging.

---

> ### Author Rebuttal · Authors · 2024-08-05
>
> We thank reviewer wRCW for the detailed feedbacks on our work. In the following, we present our responses addressing the raised concerns. Should our rebuttal effectively address the concerns, we kindly hope you can raise your score.
>
> **(Weakness 1)** We agree with you that a detailed analysis of the hyperparameters is crucial to validate the robustness of our method. Regarding the number of clusters for confounders as you suggested, **we have already presented the results in our main paper**. Please refer to Tab. 4 and Sec. 5.3.2 for details. As for the parameters, in the following Table, we compare our method to the baseline (naive ResNet-50) and Q2L (SOTA method). Our method significantly outperforms the baseline, as well as the SOTA Q2L method which has much larger model size. In particular, our causal module only adds marginal parameters, but can improve the accuracy (mAP) on COCO-stuff from 56.8 to 60.6, demonstrating the effectiveness of our causal label correlation modeling.
>
> | Decouple |	Param.	| mAP on COCO-stuff All |
> | --- | --- | --- |
> | Q2L	 |175.3M |	57.2 |
> | Baseline	 | 26.9M |	55.0 |
> | Baseline+Decouple |	41.1M |	56.8 |
> | Baseline+Decopule+Causal (Ours) |	45.3M |	60.6 |
>
> **(Weakness 2)** We agree with you that a discussion about the limitations is valuable for a more complete understanding of our method. Actually, **we have discussed the limitations of our current confounder modeling and causal intervention in Sec. A.1** (Supplemental Material). We plan to add such discussions to the main body of our paper.
>
> Regarding the confounders, **we have investigated the effects of different confounder modeling methods (see Tab. 6)**. In addition, we also add the comparisons of our K-means based confounders and random confounders, which clearly show that inaccurate confounders will lead to significant performance degradation.
>
> |Method |	Exclusive |	Co-occur |	All |
> | --- | --- | --- | ---|
> |Random |	22.1 |	66.3 |	56.1 |
> |K-means |	29.7 |	69.6 |	60.6 |
>
> **(Weakness 3)** We agree with you that comprehensive ablation study is helpful for understanding the importance and effectiveness of each component of the proposed method. Actually, **we have presented ablation experiments to validate the effectiveness of each component of our method.** Please refer to Tab 3 and Sec. 5.3.1.
>
> **(Question 1)** Thanks for suggesting this reference paper. There are two core differences between our method and the method proposed in this paper. Firstly, the meaning of the nodes in the Structural Causal Model is different: we consider two different target objects as $X$ and $Y$, while the reference paper considers the target object as $X$ and model prediction as $Y$. Secondly, our high-level considerations in confounder modeling are different. The reference paper considers co-occurring objects as confounders. While in our paper, we argue that purely modeling confounders with object-level features is insufficient, and the confounders should also include background or text information. We will add discussions about our paper and this reference paper in our revised paper.
>
> **(Question 2)** Please refer to our response to weakness 1.
>
> **(Question 3)** Please refer to our response to weakness 2.
>
> **(Question 4)** **In Sec 4.3 (line 203~217) we discuss our understanding about confounder modeling in context of multi-label image recognition.** We argue that confounders should not only consider the objects with labels defined by the dataset, but also include other types of objects, image background and even image textures. Based on this understanding, we present to model the confounders by clustering the spatial features extracted by a pre-trained classification network, which may characterize confounders with object-level, texture-level and scene-level concepts. In Tab. 6, we investigate the impact of different confounder modeling methods, where our method achieves the best result.
>
> **(Question 5)** Our method aims to suppress spurious label correlations and enhance causal label correlation. Therefore, if the correlation between two labels are spurious, our method can always suppress their label correlations for multi-label image recognition. We would like to highlight that this does not mean that our method can always make right prediction, but means that it can always suppress spurious label correlations to reduce the probability of error prediction caused by it.
>
> **(Question 6)** How to handle noisy or incomplete labels is an important research topic in machine learning. However, this is not the focus of our paper, and our current method cannot address the partial label issue. Thanks for your suggestion, and we consider extending our method to the partial label setting as our future work.
>
> **(Question 7)** We would like to highlight that pure feature decoupling using the Transformer decoder cannot reduce contextual biases. To reduce contextual biases, we explicitly model causal label correlations based on the decoupled features and confounders, with the guidance of causal intervention in causal theory.
>
> **(Limitation)** On one hand, confounder modeling is a crucial step in our method, and bad choices may lead to obvious performance degradation.  But, on the other hand, we would like to highlight, it remains an open question about confounder modeling for visual recognition tasks. In this work, we present our understanding about confounders (see line 203~217), and then develop a simple but effective modeling approach based on pre-trained classification network and K-means clustering. In our ablation experiments (Tab. 6), we show the advantages of our confounder modeling approach.
>
> In short, what we would say is that for this open question (confounder modeling), we have presented our rational analysis, as well as a simple approach with experiments validating its effectiveness.

---

> > ### Author Response · Authors · 2024-08-12
> >
> > Dear Reviewer,
> >
> > We hope that our rebuttal has effectively clarified your confusion and that the additional experiments we provided have strengthened the validity of our approach. We eagerly await your feedback on whether our response has adequately resolved your concerns, or if further clarification is needed.

---

### Official Review · Reviewer_NEe5 · 2024-07-11

**Soundness:** 3
**Presentation:** 2
**Contribution:** 3
**Rating:** 4
**Confidence:** 4

**Summary:**

This paper proposes a causal intervention mechanism for multi-label image classification, where causal label correlations are pursued and spurious label correlations are suppressed. To achieve this, the authors frame a pipeline consisting of a branch for decoupling label-specific features and a branch for summarizing causal label correlations. The results from both branches are combined for final predictions on image labels. Comparative experiments and ablation studies demonstrate the effectiveness of the proposed causal intervention mechanism.

**Strengths:**

- The paper is generally well written with clear motivation and objectives.
- Causal intervention is technically novel and well motivated in terms of multi-label image classification.
- Experimental results are impressive, outperforming sub-optimal methods by a considerable margin. Ablation studies are aslo well designed to showcase the contribution.

**Weaknesses:**

- Line 175: 'Correaltions' -> 'Correlations'.
- Line 237: 'Transformer encoder' -> 'Transformer decoder'.
- $f_{fc}$ in Eq.6 and $f_{fc}$ in Eq.11 should be different if their parameters are not shared.
- In Figure 4, in the causal label correlation branch, the confounder features are added into label-specific features. However, the outputs are not seen to be used in subsequent steps, and it seems that only the label-specific features are utilized for causal intervention. The diagram of this module needs to be improved.
- More experimental evidence should be provided to verify the effectiveness of the confounder modeling. For example: using random vectors to replace cluster centers as confounders. Only the feature visualization and ablation study on clustering center number are unconvincing.
- Although this paper is well motivated, the modeling process, especially Equation 11, is confusing.

**Questions:**

- How is the operation $f_{merge}$ removed from the second line of Eq.11?  Why is the summation over all confounders $c$ also removed? Even if it can be removed, which confounder does $c$ in the last line of the formula refer to? The Eq.11 is confusing and needs further clarification. It would be best if the authors add a pseudocode to illustrate this process.z
- Are the results in Table 1 and Table 2 reported by re-training these models on the relevant datasets? If so, the authors should clarify the experimental details for fair comparison.
- According to the Table 8, in terms of the intra-dataset comparisons on MS-COCO, Q2L achieves the same performance in mAP as the proposed method. However, Q2L only requires a Transformer decoder for decoupling label-specific features. Therefore, we question the generalizability of the proposed causal intervention mechanism on multi-label image classification task, wondering whether it is only effective on specific datasets.

**Limitations:**

See Weakness and Questions.

---

> ### Author Rebuttal · Authors · 2024-08-05
>
> We thank reviewer NEe5 for the constructive comments and suggestions. In the following, we present our responses addressing the raised concerns. Should our rebuttal effectively address the concerns, we kindly hope you can raise your score.
>
> **(Weakness 1 and 2)** Thanks for your reminding, and we will correct these typos in our revised manuscript.
>
> **(Weakness 3)** To differentiate these two different fully-connected layers, we will use $f_{fc1}$ and $f_{fc2}$ in Eq. 6 and Eq. 11, respectively.
>
> **(Weakness 4)** Thanks for your comments, which remind us that the current diagram for causal label correlation modeling in Fig. 4 lacks a crucial arrow to “feed” the added features into the cross-attention operation. We update our Fig. 4 (see our attached pdf file) to show this.
>
> **(Weakness 5)** Thanks for your kind suggestion, and we agree that random confounders should be the simplest baseline for comparison. We conduct experiments for this random version of confounders: it leads to significant performance degradation (measured by mAP), especially on the “Exclusive” subset of COCO-Stuff. We believe that confounders should be modeled at semantic level (which cannot be achieved by random vectors), and thus present our K-means based solution upon semantic features extracted by a pre-trained classification backbone network.
>
> | Method | 	Exclusive |	Co-occur	 | All |
> | --- | --- | --- | ---
> | Random	|  22.1	 |66.3	 | 56.1 |
> | K-means |	29.7 | 	69.6	 | 60.6 |
>
> **(Weakness 6)** We agree with you that our modeling process (as formulated in Eq. 11) should be further clarified. Please refer to our responses to Question 1 in the following.
>
> **(Question 1)** Regarding the $f_{merge}$ in the first two rows in Eq 11, it represents our high-level idea that “We seek for a model to combine the information of all label-specific features $x_i$ and one potential confounder feature c to predict the logit of label $Y_j$. From the third row in Eq.11, we merge all label-specific features $x_i$ into a feature matrix $X$, and thus $f_{merge}$ can be removed.
>
> Regarding the summation over all confounders, we thank for your kind reminding: it cannot be removed in Eq. 11. We will update Eq. 11 in our revised paper as the following.
>
> $$
> \begin{align}
>   {Z}\_c &= {X} + c \,, \\\\
>   \hat{y}\_{causal}^j &= f_{merge}([P(Y_j|do(X_1), ..., P(Y_j|do(X_N)]) \\\\
>               &= f_{merge}([\sigma(\sum_{c} f_{y_j}(x_1, c) \cdot P(c)), ..., \sigma(\sum_{c} f_{y_j}(x_N, c) \cdot P(c))]) \\\\
>               &\approx \sigma(\sum_{c} f_{y_j}({X}, c) \cdot P(c)) \\\\
>               &= \sigma(\sum_{c} f_{fc2}(f_{cross\\_atten}(y_j, {Z}_c, {Z}_c))\cdot P(c))
> \end{align}
> $$
> where ${Z}\_c$ is the addition-based combination of all label-specific features ${X}$ and a confounder feature $c$,
> and $f\_{fc2}$ is a fully-connected layer applied upon the cross-attention feature to obtain the logit.
>
> We provide the following pseudocode to illustrate the implementation process of Eq. 11.
>
> ---
> Define $X \in \mathbb{R}^{N \times D}$, $C \in  \mathbb{R}^{M \times D}$, $P \in  \mathbb{R}^{M}$ is the label-specific features, confounders and priors.
>
> Z = X.unsqueeze(1) + C.unsqueeze(0) # $Z \in \mathbb{R}^{N \times M \times D}$
>
> yj = X[j, :]
>
> y_causal_j = 0
>
> for c in range(M):
>
> &nbsp; &nbsp; &nbsp; &nbsp; y_causal_j += fc(cross_atten(yj, Z[:,c,:], Z[:,c,:])) * P[c]
>
> y_causal_j = y_causal_j.sigmoid()
>
> ---
>
> **(Question 2)** To ensure fair comparisons, we retrain these models based on the codes released by authors.
>
> **(Question 3)** Honestly, we present our results on MS-COCO for completeness, rather than for showing the advantage of our method. This is because MS-COCO generally satisfies the i.i.d assumption, while our method aims to improve practical multi-label recognition where the training and test images may not follow the i.i.d assumption, as the co-occurence between objects might change in testing. We follow the basic experimental settings in [1] which also aims to overcome contextual bias (but not from a causal intervention perspective), but also present our results under challenging cross-dataset setup. In this sense, our main experimental results are sufficient to validate the core contributions of this work: our method significantly outperforms existing methods under non-i.i.d settings, and can still achieve competitive results for i.i.d setting.
>
> [1] Singh, Krishna Kumar, et al. "Don't judge an object by its context: Learning to overcome contextual bias." CVPR. 2020.

---

> > ### Author Response · Authors · 2024-08-12
> >
> > Dear Reviewer,
> >
> > We hope that our rebuttal has effectively clarified your confusion and that the additional experiments we provided have strengthened the validity of our approach. We eagerly await your feedback on whether our response has adequately resolved your concerns, or if further clarification is needed.

---

> > ### Comment · Reviewer_NEe5 · 2024-08-12
> > **Response to Author Rebuttal**
> >
> > Thank you very much for your thorough response to my comments and questions. Most of my doubts are well addressed. However, I still have concerns about the effectiveness of the proposed method, given the fact that the proposed method shares the same label-specific feature learning architecture as Q2L [1], but does not show any performance improvement on MS-COCO with the additional mechanism of causal intervention. In contrast, CCD [2], IDA [3], CMLL [4] and PAT [5] have all demonstrated the effectiveness of causal intervention on this dataset. The author's explanation is not convincing.
> >
> >
> > [1] Query2Label: A Simple Transformer Way to Multi-Label Classification (arXiv 2021);
> >
> > [2] Contextual Debiasing for Visual Recognition with Causal Mechanisms (CVPR 2022);
> >
> > [3] Causality Compensated Attention for Contextual Biased Visual Recognition (ICLR 2023);
> >
> > [4] Causal multi-label learning for image classification (NN 2023);
> >
> > [5] Counterfactual Reasoning for Multi-Label Image Classification via Patching-Based Training (ICML 2024).

---

> ### Author Response · Authors · 2024-08-12
> **Response to Reviewer NEe5**
>
> Thank you for your response. **We must emphasize that our approach does not simply add causal intervention to Q2L**. Q2L includes both a Transformer Decoder and a Transformer Encoder, and uses a specially proposed ASL Loss [1]. In contrast, without considering the causal intervention branch, we only utilize the Transformer Decoder with less parameters for feature decoupling, and train our model with general multi-label classification loss.
>
> Furthermore, **Table 8 aims for performance comparison, rather than ablation study. It does not indicate that our causal intervention method gains no performance improvement on MS-COCO.** We conducted an ablation study on MS-COCO as shown in the table below (CMLL was not compared due to lack of experiments at resolution 448x448 and unavailability of code). It can be seen that our causal intervention module still provides some improvements on the MS-COCO dataset. Compared with Q2L, we achieve similar results with fewer parameters (refer to our rebuttal for Weakness 1 of Reviewer wRCW). Additionally, when compared with other methods specifically designed based on causal theory for the MS-COCO dataset, our method still achieves competitive results.
>
> | Method | mAP |
> | --- | --- |
> | Res101 | 79.1 |
> | Res101 + Decouple | 83.7 |
> | Res101+Decopule+Causal (Ours) | 84.9 |
> | Q2L | 84.9 |
> | CCD | 84.0 |
> | IDA | 84.8 |
> | PAT | 85.0 |
>
> We consider pursuing higher accuracy on MS-COCO as our future work, by combining our causal intervention method with more advanced backbone networks, multi-label loss, or pre-training methods.
>
> [1] Asymmetric loss for multi-label classification, 2020.

---

> > ### Comment · Reviewer_NEe5 · 2024-08-12
> > **Response to Author Rebuttal**
> >
> > Thank you for your quick reply. All my concerns are addressed.

---

> > > ### Author Response · Authors · 2024-08-12
> > > **Response to Reviewer NEe5**
> > >
> > > We sincerely thank you once again for the constructive comments and suggestions. Should our rebuttal effectively address the concerns, we kindly hope you can raise your score, which we believe is vital for the final decision on our work.

---

### Author Rebuttal · Authors · 2024-08-06

We thank the reviewers for their thoughtful analysis and feedbacks, which are invaluable for understanding how to improve our paper.  We address the questions and concerns raised by each reviewer point-by-point in the respective threads below.  We also attach a PDF containing one updated Figure in response to Reviewer **NEe5,** and one additional Figure ****for Reviewer **BNe7.**

In general, reviewers agree that we present a novel effective pipeline that integrates causal label correlation modeling that can improve practical multi-label recognition where the training and test images may not follow the i.i.d assumption, as the co-occurence between objects might change in testing.  Specifically, we appreciate their assessment of our paper writing as **“well written with clear motivation and objectives”** (NEe5) and **“well-written and easy to understand”** (BNe7). The reviewers concur that **“Causal intervention is technically novel and well motivated in terms of multi-label image classification”** (NEe5). They give positive comments on our experiments that **“Experimental results are impressive, outperforming sub-optimal methods by a considerable margin. Ablation studies are also well-designed to showcase the contribution”** (NEe5), **“extensive experiments across multiple datasets, demonstrating the effectiveness of the proposed method. The results consistently show improvements over existing approaches, particularly in scenarios with contextual biases, underscoring the practical value of the method”** (wRCW), and **“This method seems easy to implement. The approach achieves good results. The problem is interesting in multi-label recognition tasks”** (BNe7).

Most of the reviewers expect more in-depth analysis and comparisons of the **confounder modeling** in our pipeline. Here, we would like te state the motivation, approach, validation, and limitation of our confounder modeling in detail.

**Motivation**: On one hand, confounder modeling is crucial for causal intervention, as evidenced by our new experiments that compare random confounders and our method (see our response to Weakness 5 for reviewer **NEe5**). But on the other hand, based on our survey, it remains an open question about cofounder modeling for visual recognition tasks. In this work, we present our understanding about cofounders modeling. That is, confounders for recognizing a certain object are often hard to define and enumerate – objects, scene, and even the texture of the environment are all potential confounders.

**Approach**: We model the confounders by clustering the spatial features extracted with a pre-trained classification network, as the classification network tends to activate object-level, texture-level, or scene-level semantic concepts [1]. By clustering spatial features with K-means, we obtain a compact set of prototypes to represent potential confounders like objects, textures and scenes. Our approach is significant different from previous works like VC R-CNN [2] that relies on bounding box annotations (which are often absent for image classification task), and only considers pre-defined objects as confounders.

**Validation**: In the main body of our paper, we validate the effectiveness of our confounder modeling approach in Tab. 6, and investigate the impact of the number of clustering centers in Tab. 4. Furthermore, we compare our confounder modeling approach with simple random confounders during rebuttal (response to Weakness 5 for reviewer **NEe5**).

**Limitation**: As described in Sec. A.1 in our Supplemental Material, our confounder modeling approach is dataset-dependent, although our cross-dataset multi-label image classification experiments can justify its generalization ability to a large extent. On the other hand, although our modeled confounders are often concerned with object-level, texture-level and scene-level concepts, it is difficult to determine the specific semantics of these concepts, resulting in limited interpretability. We leave better confounder modeling approach as our future work.

[1] Zhou, Bolei, et al. "Learning deep features for discriminative localization." CVPR. 2016.

[2] Wang, Tan, et al. "Visual commonsense R-CNN." CVPR. 2020.

---

### Decision · Program_Chairs · 2024-09-25

**Decision:**

Accept (poster)

**Comment:**

After the rebuttal and the authors-reviewers discussion period, the paper got 2 borderline rejects and 1 strong accept. Overall, the reviewers think the proposed method is novel and interesting. The proposed method uses a causal intervention approach to identify the causal label correlations and suppress spurious label correlations. The rebuttal answered most of the concerns about the experimental model analysis. After reading the paper, the reviews, and the rebuttal, the AC recommends to accept this paper. The AC acknowledges that some parts of the experimental section could be improved by adding the new results from the rebuttal, but the overall technical contribution is above the acceptance bar.  The AC strong encourages the authors to update the paper by adding the key results from the rebuttal.